# Diamide-based screening method for the isolation of improved oxidative stress tolerance phenotypes in *Bacillus* mutant libraries

Jonathan Walgraeve,[1] Borja Ferrero-Bordera,[2] Sandra Maaß,[2] Dörte Becher,[2] Ruth Schwerdtfeger,[1] Jan Maarten van Dijl,[3] Michael Seefried[1]

**ABSTRACT**  The bacterium *Bacillus subtilis* is of high importance both as a model organism for Gram-positive bacteria and as an industrial workhorse in the production of biomolecules. In recent years, advancements have been made to engineer the bacterium even further toward industrial applications. In this study, we present a novel screening method for mutant libraries using diamide, an oxidizing agent that binds free thiols and creates disulfide bonds between them, thereby causing a so-called "disulfide stress" in bacteria. The method shows promise to selectively identify phenotypes in *B. subtilis* with improved tolerance toward oxidative and disulfide-associated stress. Phenotypes initially identified by transposon mutagenesis were recreated through targeted gene deletions. Among the resulting deletion mutants, the largest difference in diamide tolerance compared to the parental strain was observed for *pfkA* and *ribT* deletion strains. A proteomics analysis showed that diamide tolerance can be achieved through different routes involving increased expression of stress management proteins and reduced availability or activity of the RNA degradosome. We conclude that our screening method allows the facile identification of *Bacillus* strains with improved oxidative stress tolerance phenotypes.

**IMPORTANCE**  During their life cycle, bacteria are exposed to a range of different stresses that need to be managed appropriately in order to ensure their growth and viability. This applies not only to bacteria in their natural habitats but also to bacteria employed in biotechnological production processes. Oxidative stress is one of these stresses that may originate either from bacterial metabolism or external factors. In biotechnological settings, it is of critical importance that production strains are resistant to oxidative stresses. Accordingly, this also applies to the major industrial cell factory *Bacillus subtilis*. In the present study, we, therefore, developed a screen for *B. subtilis* strains with enhanced oxidative stress tolerance. The results show that our approach is feasible and time-, space-, and resource-efficient. We, therefore, anticipate that it will enhance the development of more robust industrial production strains with improved robustness under conditions of oxidative stress.

**KEYWORDS**  *Bacillus subtilis*, mutagenesis, library screening, diamide, disulfide

Address correspondence to Michael Seefried, Michael.Seefried@abenzymes.com.

J.W., R.S., and M.S. are employees of AB Enzymes GmbH.

See the funding table on p. 19.

The Gram-positive bacterium *Bacillus subtilis* has been found in a wide variety of biotopes and has been studied extensively in the context of both academic and industrial interests (1). It is the prototype bacterium for the phylum *Firmicutes*, a model for Gram-positive bacteria, and was one of the first organisms to have its genome sequenced (2). In recent years, *B. subtilis* has gained increasing importance for the production of proteins (3) and metabolites such as surfactin and riboflavin (4–7). For these purposes, production strains are being tailored to show desirable growth properties, high productivity, and improved stress resistance (8). One such strain quality

is the resistance against oxidative stress that is encountered during cellular growth (9). Oxidative stress can originate from a variety of sources and is caused by oxidative damage of cellular components (10). An example of this is the oxidation of free thiol groups of cysteine residues in proteins (11). Molecules that can cause this type of stress are reactive oxygen, nitrogen, chlorine, and electrophilic species. The oxidative stress response generally refers to the adaptations made by the cell to deal with the adverse effects caused by these oxidants (12). The specific bacterial responses differ depending on the stress agent involved although overlap exists (10, 12).

Recent investigations showed that genome-minimized strains of *B. subtilis* have improved production traits over strains with the complete genome (13–15). While these mini- and midi-*Bacillus* strains show, for example, significantly higher enzyme production, this comes with longer fermentation times and reduced medium utilization (16, 17). The latter two traits are undesirable in the context of industrial applications. However, the available data showed that, in principle, the strategic removal of unnecessary genes is a good approach to improve strain qualities.

Transposon mutagenesis is commonly employed to generate phenotypic variations and identify their genetic basis (18–20). The resulting insertion can result in a gene disruption, which can be mapped with little issue due to the genome and transposon sequences being known. As transposon mutagenesis allows for the creation of large and varied mutant libraries, the subsequent challenge lies in narrowing down the number of candidate mutations with beneficial traits by screening in a time-, space-, and resource-efficient manner.

In the present study, a method is described for easy and fast screening of large libraries of (transposon) mutants for those bacteria that show enhanced tolerance toward oxidative stress. The method allows for easy and fast screening, covering the genome multiple times, and is based on the use of tetramethylazodicarboxamide (diamide). This reagent induces oxidative and disulfide stress during the cultivation of bacteria (21). Using diamide, Martin et al. previously showed the beneficial effects of *mfD* overexpression to achieve oxidative stress tolerance in *B. subtilis* (22). The mode of action of diamide is to bind free thiols and create disulfide bonds between them. These interactions also occur between free cysteine residues of cellular proteins and can be inter- or intramolecular. Most commonly though, S-thiolations with low molecular weight (LMW) thiols, such as bacillithiol, coenzyme A (coA-SH), and cysteine, were observed (23). This type of thiolation is reversible and is regarded as a bacterial strategy to protect its thiol-containing cellular constituents during conditions of high oxidative stress (24). When present in sufficient concentrations, diamide causes bacterial growth arrest. As a diamide molecule reacts with free thiols or degrades due to heat or light, it is converted to hydrazine, which cannot interact with thiols. Once all the diamide has been sufficiently cleared from the environment, bacterial growth will resume (21).

The present study was aimed at developing a diamide-based screening method for *Bacillus* strains with improved oxidative stress tolerance. The method allowed to screen an extensive library of *B. subtilis* transposon mutants for increased diamide tolerance. Subsequently, the transposon insertion loci were mapped, and the causality of the transposon mutations and their phenotypes was verified by seamless gene deletion. Lastly, a more in-depth analysis was performed on two deletion mutants with the highest oxidative stress tolerance, using proteomic analyses.

## MATERIALS AND METHODS

### Strains and plasmids

The *B. subtilis* strain DB430 Δ*lipA* was used during this study, both as a parental strain for transposon mutagenesis and as baseline control/reference strain during mutant screening and characterization (25). This strain has reduced exo-protease activity due to the deletion of genes for three secreted proteases and one membrane-associated protease. In addition, it lacks the gene for the secreted esterase LipA. The *B. subtilis*

SCK6 strain, containing a xylose-inducible competence cassette, was used for plasmid construction purposes (26).

For transposon mutagenesis, the pMarB plasmid was used as described by Le Breton et al. (27). It carries an erythromycin-resistance marker on the vector part and a kanamycin-resistance marker in the transposon region. A SigB-regulated HimarI transposase is encoded by the plasmid to initiate the movement of the transposon.

For targeted gene deletions, the pKVM2 seamless gene deletion vector was used as described by Rachinger et al. (28). The plasmid carries a tetracycline-resistance marker as well as a β-galactosidase gene, both of which are used for selection.

## Culture media and reagents

Bacteria were cultured in lysogeny broth (LB) prepared from pre-mixed powder (25 g/L, LB Lennox, Roth). During cultivation, the appropriate antibiotics or other additives were included from prepared stocks. LB agar plates were made using 40 g/L LB agar (Luria-Miller, Roth). Media were prepared using demineralized water and autoclaved at 121°C for 20 min. Working concentrations of the following antibiotics were used: 2 µg/mL erythromycin (Roth), 10 µg/mL kanamycin-B (Roth), and 10 µg/mL tetracycline (Roth). Diamide was used at concentrations of 0.05–14 mM (Sigma-Aldrich).

Lysis buffer for gDNA extraction was prepared in demineralized water and holds 20 mM Tris-HCl pH 8.0, 2 mM EDTA pH 8.0, 1.2% Triton X-100, and 20 mg/mL lysozyme. HEPES buffer 1 M stock solution was prepared in demineralized water. The pH was adjusted to pH 7.0 using NaOH. The stock solution was stored at 4°C. Working solutions were made fresh each time by diluting them to 20 mM with demineralized water.

## Kinetic diamide growth assay

Overnight cultures were prepared in a 96-deep well (2 mL) plate (Greiner Bio-one). To this end, the wells were filled with 1 mL fresh LB medium and inoculated from a plate with picked transposon mutants. In addition, three wells were inoculated with the parental DB430 Δ*lipA* strain, which was included as a control. The plate was then sealed with a semi-permeable membrane (Rayon film, VWR), and the bacteria were grown overnight at 37°C and 800 rpm orbital shaking in an incubator (Infors HT Multitron II). The following day a sterile transparent 96-well microtiter plate (Greiner Bio-One) was prepared with 200 µL LB medium containing 2 mM diamide in each well. The wells were then inoculated with 10 µL of overnight culture from the corresponding well from the 96-deep-well plate. The microtiter plate was placed in a plate reader (BioTek Synergy Mx) preheated at 37°C. The plate was incubated under constant shaking and the optical density at 600 nm ($OD_{600}$) was measured at 10-min intervals for the duration of 30 h. The deep-well plates with the overnight cultures were frozen at −80°C after inoculating the microtiter plate for the kinetic growth assay.

## DNA methods

### Transposon mutagenesis

An overnight culture in a baffled 100 mL shake flask containing 10 mL LB medium with 2 µg/mL erythromycin was inoculated from the plate with *B. subtilis* DB430 Δ*lipA* carrying the pMarB plasmid. The culture was grown at 28°C at 180 rpm orbital shaking in an incubator (Infors HT Multitron II). The following day, the overnight culture was used to inoculate 10 mL LB without antibiotic $OD_{600}$ of 0.1, and culturing in a 100 mL baffled shake flask was continued at 37°C and 180 rpm orbital shaking for 5 h to dilute the plasmid copy number in the cells. The cells were then subjected to a 30-min heat shock in a 50°C water bath to induce the SigB-dependent expression of the HimarI transposase.

Cells were subsequently plated (100 µL undiluted cell suspension per plate) on LB agar plates containing 200 µM diamide and 10 µg/mL kanamycin. In addition, a dilution series was prepared and plated on LB agar plates containing 10 µg/mL kanamycin

followed by counting of colony-forming units (CFUs) the following day. All plates were incubated at 37°C.

## gDNA extraction

The deep-well plates containing the overnight cultures for the kinetic growth assay were taken from the −80°C storage and thawed. Next, the broth with the selected transposon mutants was transferred to individual 1.5 mL microcentrifuge tubes. The bacteria were then collected by 1-min centrifugation at 13,000 rpm in a tabletop centrifuge after which the supernatant was discarded. The bacterial pellet was resuspended in 180 µL of lysis buffer. A total of 4 µL of RNase A (100 mg/mL) was added, and the tubes were incubated for 30 min at 37°C. Next, the purification of gDNA was performed according to the Qiagen Blood & Tissue kit (Qiagen Cat. No. 69506) protocol. The eluted DNA samples were stored at −20°C until use.

## Polymerase chain reaction

All primers used for PCR in this study are listed in Table S1.

Fusion primer and nested integrated PCR (FPNI-PCR) were used as described in reference (29). The method produces a fragment containing part of the transposon and the genomic locus where it has been inserted. The FPNI-PCR was checked using gel electrophoresis with a 1% agarose gel in Tris-acetate EDTA (TAE) buffer (40 mM Tris base, 20 mM acetic acid, 2 mM EDTA) containing Roti-Safe Gelstain (Roth). The gel was run for 30 min at 8 V/cm after which the bands were checked using UV light. Amplified DNA fragments from successful FPNI-PCRs were then purified using spin columns (Wizard SV Gel and PCR Clean-up system, Promega). From the purified PCR product, 12 µL was mixed with 3 µL sequencing primer (10 µM) and sent for Sanger sequencing (Microsynth GmbH, Göttingen).

## Seamless gene deletion

Targeted gene knockouts were made using a plasmid-based seamless deletion system. The entire gene locus was removed, leaving the regulatory sequences intact so as not to disrupt any operons. The regions surrounding the gene locus were amplified using Phusion PCR and cloned into the pKVM2 deletion vector using a Gibson assembly method (NEBuilder HiFi DNA assembly). The product was subsequently introduced into induced competent *B. subtilis* SCK6 cells, which were plated on LB agar containing tetracycline (10 µg/mL). Plates were incubated at 28°C to allow replication of the temperature-sensitive plasmid in transformed *B. subtilis* SK6 bacteria. The correct construction of vectors for gene deletion was checked by colony PCR and Sanger sequencing of isolated plasmids. Correct plasmids were used to transform the target strain *B. subtilis* DB430 Δ*lipA* using natural competence (30). Integration of plasmids into the bacterial chromosome was achieved by cultivation at 42°C, where no plasmid replication is possible and subsequent plating under antibiotic selection. Integration of the plasmid was confirmed by colony PCR. The plasmid was excised from the genome upon selection-free cultivation, where the bacteria can either revert to the wild-type genotype or the desired deletion genotype. Identification of correct gene deletions was performed using gene-specific primers outside of the locus.

## Proteomics and redox state determination

Strains were streaked on LB plates and grown at 37°C overnight. Baffled 125 mL shake flasks containing 10 mL of LB medium were inoculated in triplicate from the plates and grown overnight at 37°C at 180 rpm orbital shaking in an incubator (Infors HT Multitron II). The following day, the $OD_{600}$ was measured by diluting the overnight cultures 1:10 in a fresh LB medium. From the overnight cultures, baffled 250 mL shake flasks containing 20 mL LB medium were inoculated to a starting $OD_{600}$ of 0.1. These shake flasks were then placed in shakers (180 rpm orbital shaking) in a climatized room at 37°C. When the

$OD_{600}$ of the cultures reached 1.0, a sterile filtered (0.22 µm) diamide solution in LB was added to a final concentration of 2 mM diamide in the culture. To negative controls, the same amount of LB was added.

Samples were then harvested at 30 min, 1 h, and 4 h after diamide addition. For each sampling point and for each strain, three diamide-treated replicates and three non-treated replicates were harvested. The $OD_{600}$ was measured, and the cell suspension was transferred to 15 mL centrifugation tubes. The bacteria were collected by centrifugation at 4,400 rpm for 5 min in a cooled 4°C centrifuge (Thermo Scientific Heraeus Multifuge 3S-R). The supernatant was discarded, and the cell pellet was washed with 1 mL of 20 mM HEPES pH 7.0 buffer. The cell pellets were then flash-frozen in liquid nitrogen after which they were stored at −80°C until further analysis.

### Proteomics sample preparation and LC-MS/MS measurements

Cell pellets were resuspended in TE Buffer (10 mM Tris, 1 mM EDTA, pH 7.5) and mechanically disrupted using a FastPrep24 (MP Biomedicals, three cycles, 30 s each, maximum acceleration). The resulting extract was centrifuged for 15 min at 14,400 $g$ at room temperature and protein-containing supernatant fractions were recovered for further sample preparation. For samples corresponding to 30 min after the diamide stress onset, differential cysteine labeling was performed as described in reference (31). Briefly, natively reduced cysteines were first blocked upon disruption with standard iodoacetamide (Sigma-Aldrich). Thereafter, reversibly oxidized cysteines were reduced with tris(2-carboxyethyl)phosphine (TCEP; Sigma-Aldrich) and subsequently labeled with heavy iodoacetamide-$^{13}C_2$, 2-$d_2$ (Sigma-Aldrich). The labeling order was switched for the third biological replicate of each cultivation in order to exclude potentially marginal effects that could be caused by the different labels.

The protein concentration of the resulting protein extracts was determined using a bicinchoninic acid (BCA) assay (Pierce BCA Protein Assay Kit, ThermoFisher Scientific). For all samples, 20 µg protein was tryptically digested using the S-Trap protocol according to the manufacturer (Protifi). The peptide concentration was determined using the Pierce quantitative fluorometric peptide assay. For liquid chromatography with tandem mass spectrometry (LC-MS/MS), 2 µg of peptide mixture per biological replicate was desalted using U-C18 Zip Tips (Merck).

Peptide mixtures (1 µg) were separated on an Easy nLC 1000 coupled online to an Orbitrap Velos mass spectrometer (ThermoFisher Scientific). In-house self-packed columns (i.d. 100 µm, o.d. 360 µm, length 200 mm) packed with 3.0 µm Dr. Maisch Reprosil C18 reversed-phase material (ReproSil-Pur 120 C18-AQ) were loaded with 18 µL of buffer A [0.1% (vol/vol) acetic acid] at a maximum pressure of 220 bar. Peptide elution was performed in a 180 min non-linear gradient with buffer B [0.1% (vol/vol) acetic acid in 95% (vol/vol) acetonitrile] at a constant flow rate of 300 nL/min. Eluted peptides were measured in the Orbitrap with a resolution of R = 30,000 with lock-mass correction activated. Following each MS-full scan, up to 20 dependent scans were performed in the linear ion trap after collision-induced dissociation fragmentation based on the precursor intensity.

### Data processing for relative proteomics

Raw files were imported into MaxQuant (2.2.0.0) for data processing and protein identification. Protein database searches were performed against a forward reverse *B. subtilis* 168 database (UP000001570) with common contaminants added by MaxQuant with the following parameters: peptide tolerance, 4.5 ppm; min fragment ions match per peptide, 2; primary digest reagent, trypsin; missed cleavages, 2; variable modifications, carbamidomethyl C (+57.0215), heavy carbamidomethyl C (+61.04072, for samples with differential cysteine labeling only), oxidation M (+15.9949), and acetylation N, K (+42.0106). Results were filtered for a 1% false discovery rate (FDR) on spectrum, peptide, and protein levels. The prerequisite for protein identification was a minimum of two peptides.

Processed data were analyzed using Python 3.9. Numpy and Pandas libraries were used for data importation and cleaning coupled to an in-house pipeline. Normalized label-free quantitation (LFQ) was used for relative quantification of the identified proteins with a minimum of two valid values from three experimental replicates per condition. Scipy and Statsmodels packages were used for statistical analysis of the quantified proteins. Fold changes were calculated from averaged log2-transformed LFQ intensities and $t$-tested for significance. The resulting $P$-values were corrected using FDR correction (alpha = 0.01). Significance was considered for log2-transformed fold change (log2FC) > log2(1.5) ($\approx 0.58$) and adjusted $P$-value (adj. $P$) < 0.05.

For redox state determination, intensity ratios for one cysteine-containing peptides were calculated and tested through unpaired $t$-test. $P$-values < 0.05 were considered significant (31).

## RESULTS

### Plate screening

To screen for mutant strains with improved oxidative stress resistance, first, a suitable diamide concentration range had to be determined. The criteria for this range required a selection pressure that was both highly selective and time-efficient.

In parallel, a transposon mutant library was generated based on the *B. subtilis* DB430 Δ*lipA* strain carrying pMarB. Then, a dilution series of bacteria from this library was prepared and plated on LB agar containing 10 µg/mL kanamycin and diamide in the range of 0–400 µM. The plates were incubated at 37°C until growth was observed (Fig. 1).

No bacterial growth could be detected on plates containing 400 µM diamide within 40 h. It is possible that colonies would eventually show after longer incubation, but this would fall outside the practical time window, and it would increase the chance of accumulating undesired spontaneous mutations that are not related to a transposon insertion (32). It was concluded that a concentration of 200 µM diamide was most appropriate for screening mutant bacteria with increased diamide tolerance. At this concentration, growth was sufficiently inhibited, even when using undiluted heat-shocked cell suspensions. Moreover, the application of 200 µM diamide allowed the selection of single colonies. A lower concentration of 100 µM diamide only allowed selection of individual colonies for up to 16 h, but soon thereafter satellite colonies would appear with limited diamide tolerance as this diamide concentration was not sufficiently selective.

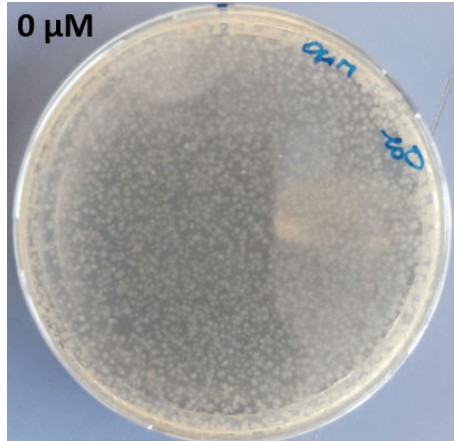 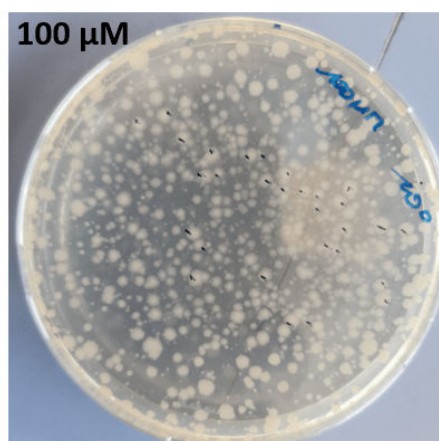 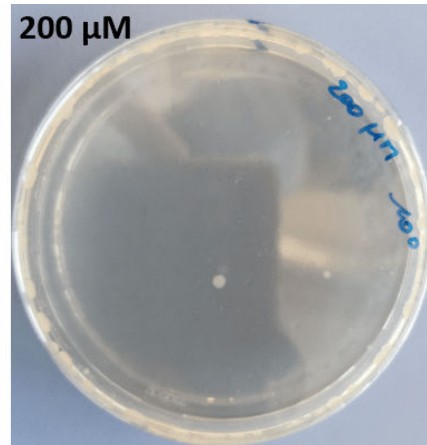

**FIG 1** LB agar plates with differing diamide concentrations onto which a 100 µL suspension of heat-shocked *B. subtilis* DB430 Δ*lipA* carrying pMarB (OD$_{600}$ ≈ 3) was plated. Note that the colony density (varying from fully overgrown to defined single colonies) depended on the diamide concentration (left to right: 0, 100, and 200 µM) and incubation time (40 h in these figures).

## Library generation and coverage

Mutant libraries were generated by four cycles of transposon mutagenesis, plated on LB agar containing 10 µg/mL kanamycin and 200 µM diamide, and incubated at 37°C. Colonies were picked at 16, 24, and 40 h after plating and transferred to fresh LB-kanamycin plates. Thus, a total of 312 candidate colonies of bacteria with potentially increased diamide tolerance were selected. In parallel to the selection of diamide-tolerant transposon mutants, CFUs were counted on LB plates with kanamycin, but without diamide, in order to estimate the total number of bacteria with transposon mutations (Table S2). These plates were incubated at 37°C to avoid plasmid replication. The results showed that an average viable count of $1.4 \pm 0.7 \times 10^5$ CFU/mL was obtained after heat shock. With 100 µL aliquots plated on 97 plates, this implied that a total of about $1.4 \pm 0.7 \times 10^6$ viable transposition events were screened for diamide tolerance.

As determined using the PePPer webserver, the *B. subtilis* 168 genome has a total number of 561,325 TA sites available for transposon integration (33). The number of mutants screened in our library thus represented a theoretical 2.49-fold coverage of each possible transposon insertion site in the *B. subtilis* genome. On the gene level, the coverage would be over 300-fold. Both calculations are still conservative, as they do not consider insertion sites causing deleterious mutations in essential genes.

## Microtiter plate kinetic growth assay

To distinguish between false positives, satellite colonies, and the desired diamide-tolerant mutants, a second screening was performed in microtiter plates. A diamide concentration of 2 mM in LB broth culture was found to be sufficiently inhibitory to show clear differences in resistance (Fig. S1 and S2). This finding was consistent with the findings reported in a different study, showing that overexpression of the translation-coupling repair factor Mfd reduced the growth inhibitory effect of diamide in *B. subtilis* (22). Here, the authors found the same concentration of diamide to be selective in liquid media.

Substantial differences were observed between the times at which different transposon mutants resumed growth in the presence of diamide, with some mutants showing no growth at all. To enumerate the time point at which growth resumed, the time point was taken where the $OD_{600}$ measurement exceeded 0.5. This allowed a comparison with the parental strain that was grown in the same plate (Fig. 2). This revealed that most mutants displayed a similar growth behavior as the parental strain. Yet, a number of mutants showed much shorter growth delays, and these were designated as potential diamide-tolerant candidates. To allow comparison of results obtained from different microtiter plates, the relative growth delay compared to the parental strain was calculated. The mutants that displayed the shortest relative growth delay in the presence of diamide (~1–15 h) were then selected for further analysis (Fig. 2).

## Transposon insertion locus identification

To investigate which transposon insertions had caused high tolerance to diamide, a total of 81 mutants were chosen, and their genomic DNA was extracted. Next, the insertion sites of the transposon were determined using FPNI-PCR. A PCR product that could be sequenced was recovered for 40 of these mutants. Upon analysis, 24 transposon insertions within coding regions were identified (Table 1). Other insertions were either outside of a coding region or sequencing proved unsuccessful. Interestingly, several transposon mutations were identified multiple times in the analysis. These included insertions in *pfkA* (6×), *cysE* (4×), and the 23S RNA genes (3×). The latter genes are present in multiple copies in the genome of *B. subtilis*, and their importance should thus be reconsidered accordingly. Transposon insertions outside of coding regions, while also of interest, were considered outside the scope of the present study.

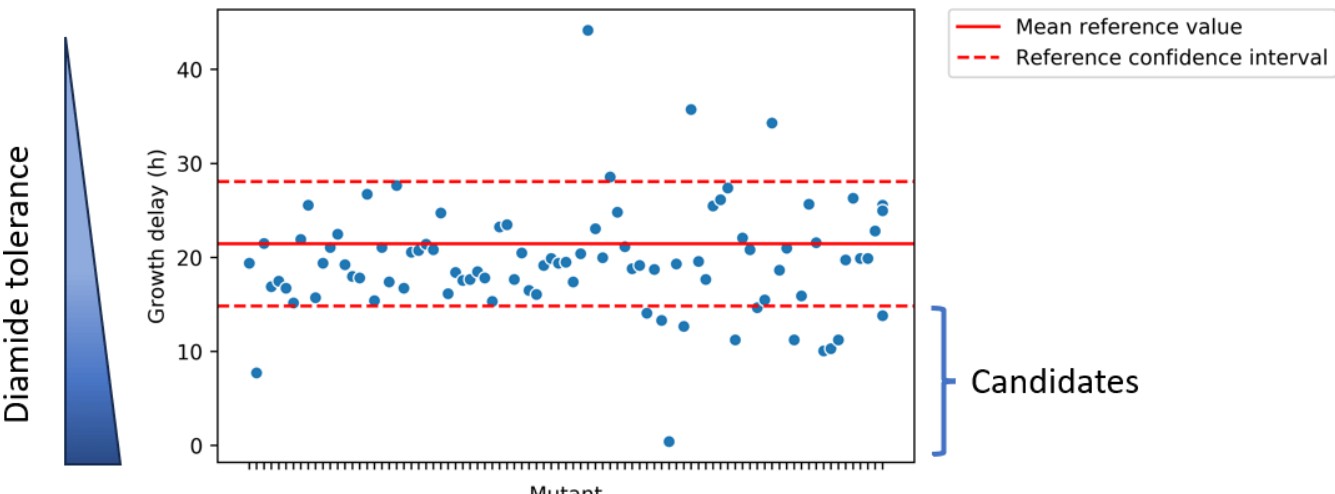

**FIG 2** Example of the selection of candidate transposon mutants with increased diamide tolerance by microtiter plate screening. Diamide tolerance was measured as the time in hours required to reach exponential growth in LB broth with 2 mM diamide. The solid and dashed red lines indicate the mean and standard deviation, respectively, as measured for the parental strain *B. subtilis* DB430 Δ*lipA* that was included as an internal control. Mutants with a growth delay below the lower dashed line were considered potentially diamide-tolerant candidates. The mutant showing the shortest growth delay was subsequently shown to carry a transposon insertion in the *pchR* gene.

## Verification of the impact of transposon insertions by gene deletions

To exclude the possibility that the selected phenotypes could be due to an unidentified secondary transposon insertion or spontaneous mutation (32), the most promising transposon mutations (Table 2) were recreated using a seamless gene deletion strategy. The target genes were selected based on the high diamide tolerance resulting from their transposon inactivation and multiple identifications of the respective transposon insertions.

Deletion of the entire *cysE* gene locus was not successful. Upon further investigation, it was observed that the transposon had been inserted in the three identified mutants at, respectively, 132, 137, and 486 bp after bp 1 of the *cysE* gene. This may have resulted in the production of a truncated protein with biological activity, or the transposition may have caused polar effects. Deletion of the *cysE* gene was previously

**TABLE 1** Identified transposon insertions inside coding regions and the respective functions according to Uniprot[a]

| Gene | Biological process | #Cysteines | #Hits |
| --- | --- | --- | --- |
| *pfkA* | Glycolysis pathway | 3 | 6 |
| *cysE* | Cysteine pathway | 0 | 4 |
| *rny* | mRNA processing | 1 | 1 |
| *pchR* | Pulcherriminic acid biosynthesis regulator | 1 | 1 |
| *bshC* | Bacillithiol biosynthesis | 2 | 1 |
| *ribT* | Riboflavin biosynthesis | 1 | 1 |
| *tyrZ* | Protein synthesis | 0 | 1 |
| *lysP* | Amino acid transport | 0 | 1 |
| *ptsI* | Phosphorylation | 2 | 1 |
| *23S RNA* | Protein synthesis | NA | 3 |
| *cgoX* | Heme biosynthetic process | 4 | 1 |
| *fabF* | Fatty acid biosynthetic process | 3 | 1 |
| *radA* | Recombinational repair | 6 | 1 |
| *yvlC* | Unknown | 0 | 1 |

[a]The number of cysteine residues in the corresponding protein is shown to pinpoint potential target sites for diamide. #Hits indicates how many times transposon insertions were identified in a particular gene.

**TABLE 2** Selected target genes for seamless deletion and reason for their selection

| Selected target | Reason |
| --- | --- |
| *pfkA* | Number of identifications (6×) |
| *cysE* | Number of identifications (4×) |
| *pchR* | Highly enhanced diamide tolerance (−64%) |
| *ribT* | Highly enhanced diamide tolerance (−38%) |
| *bshC* | Highly enhanced diamide tolerance (−29%) |
| *rny* | Highly enhanced diamide tolerance (−29%) |

shown to significantly hamper growth which could also explain difficulties in creating a seamless deletion strain (34). Further investigation of this gene locus would be required to determine its link to diamide tolerance.

The generated deletion strains were grown in quadruple in LB in microtiter plates, as was done during the screening of transposon mutants, using diamide concentrations of 0 mM, 0.5 mM, 1 mM, or 2 mM with $OD_{600}$ measurements at 10-min intervals over 48 h. Again, the time to reach $OD_{600}$ 0.5 was used to compare phenotypes (Fig. 3).

Differences in the growth of the different deletion mutants and the parental strain were negligible in the absence of diamide. However, they became evident at increasing concentrations of diamide. At a concentration of 0.5 mM, a first indication of the reduced growth delay could already be noticed. At 2 mM diamide, a distinct difference in growth behavior was observed. For strains with *rny* or *pchR* deletion, the improvement was relatively small (paired *t*-test, $P < 0.1$). However, for the Δ*ribT*, Δ*bshC*, and Δ*pfkA* deletion strains the difference was significant (paired *t*-test, $P < 0.05$). The *pfkA* deletion strain specifically showed the largest diamide tolerance compared to the parental strain. This

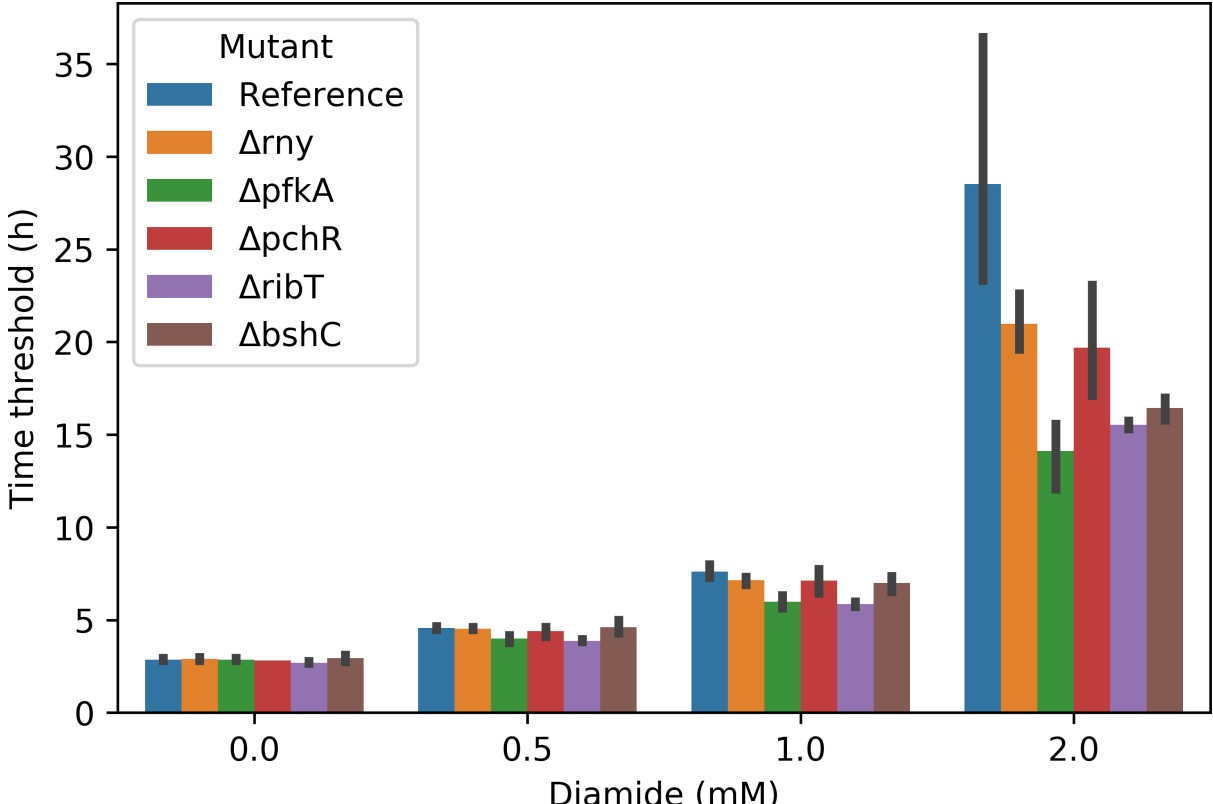

**FIG 3** Bar diagram showing the time to reach the exponential phase ($OD_{600} > 0.5$) for different deletion mutants in microtiter plate in the presence of various concentrations of diamide compared to the parental strain (reference). A reduction in this time compared to the parental strain implies that the phenotype of the respective transposon mutant is related to inactivation of the gene identified by FPNI-PCR and not to unidentified off-target effects.

particular gene was also identified in six independently selected transposon mutants, implying a strong link with improved diamide tolerance.

A follow-up experiment was performed to test the upper limits of diamide tolerance of the selected mutants. Leichert et al. reported previously that a diamide concentration of up to 2 mM would lead to growth inhibition, while a concentration of 10 mM would lead to cell lysis (21). This approach was adapted to microtiter plates (instead of shake flasks as in the original paper). Diamide was added after 3 h of incubation, when the $OD_{600}$ was around 0.5 for all replicates, to final concentrations in the range of 0–14 mM, and incubation was continued. The respective $OD_{600}$ measurements are shown in Fig. 4.

Notably, the growth delay caused by diamide depended on the cell density of the culture, similar to what was observed upon plating. Consequently, the growth delays observed at a concentration of 2 mM were very short (<30 min). At a concentration of 4 mM, bacterial lysis occurred, as evidenced by a drop in the optical density of all mutants except the *ribT* deletion mutant. This mutant went through a relatively short period of growth stasis before resuming exponential growth. The other strains were still able to resume growth after the lysis period and then, no significant differences in their growth behavior could be detected, suggesting that the effective diamide concentration

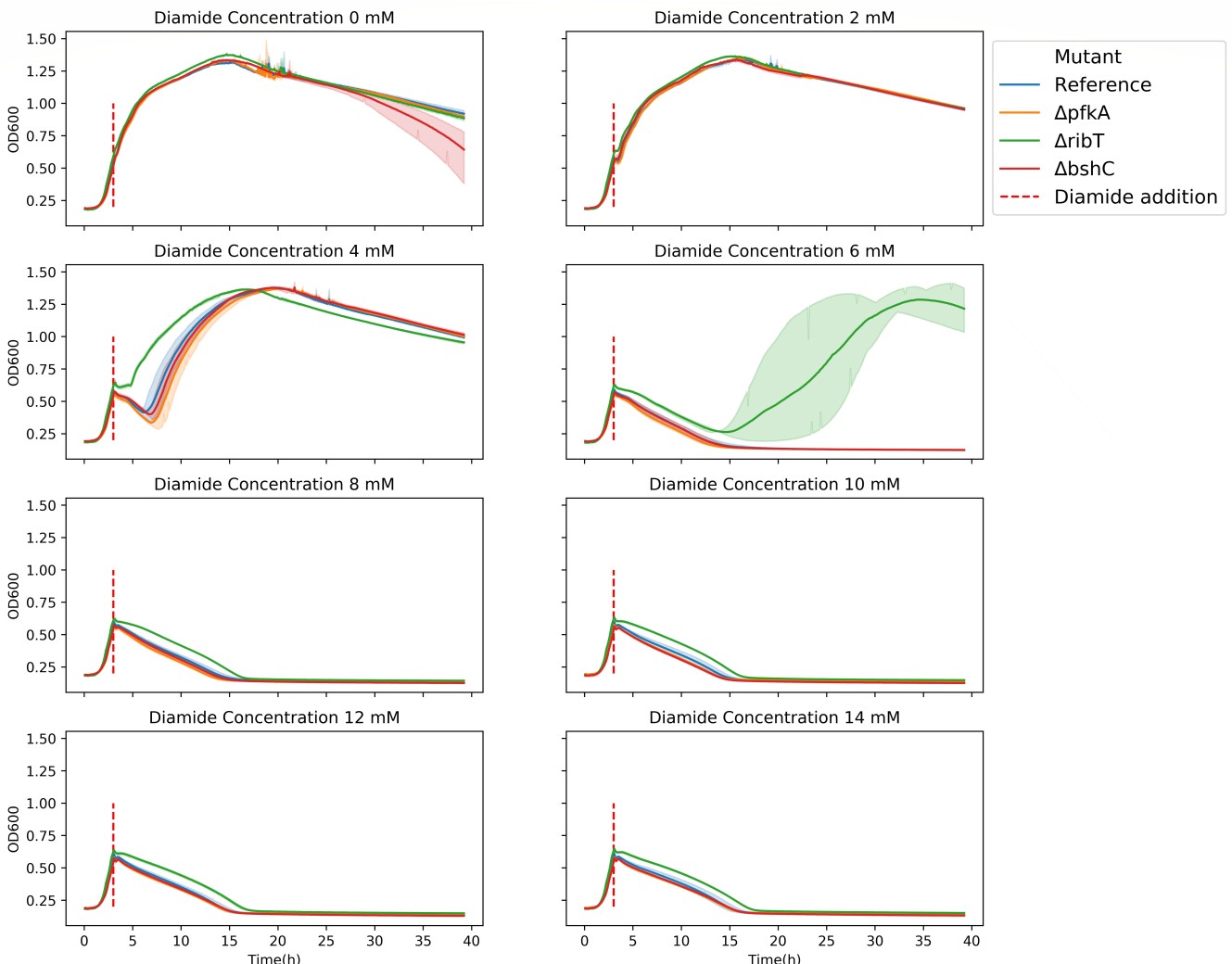

**FIG 4** Growth curves measured for microtiter plate cultures of the parental strain (DB430 Δ*lipA*, reference) and the Δ*pfkA*, Δ*ribT*, or Δ*bshC* deletion strains at various diamide concentrations. Diamide was added during the exponential phase of growth (dashed red line) to test tolerance against higher concentrations compared to the parental strain. In the plot depicting the growth of the *ribT* mutant at 6 mM diamide, the area shaded in green marks the variation in the individual growth curves.

had become too low or that the strains had adapted. When the mutants were treated with a diamide concentration of 6 mM, only the *ribT* deletion mutant was able to recover after a long period of lysis, whereas none of the other strains was able to recover. All bacteria treated with diamide concentrations of 8 mM and higher, lysed completely and did not recover at any point during the experiment.

Together, the results in Fig. 3 and 4 showed that the *pfkA* deletion provided a stronger growth advantage at lower diamide concentrations compared to the other tested mutants. The *ribT* deletion seemed to provide a more moderate protection which, however, seemed to hold at higher diamide concentrations, which the *pfkA* deletion did not show. Once the diamide concentration became too high, any advantage provided by these gene deletions seemed to be forfeited.

## Proteomics and redox state determination

The two mutants with the highest diamide tolerance, the Δ*pfkA* and Δ*ribT* mutants, were used for further investigation to obtain deeper insights into the adaptations underlying their diamide tolerance. The parental strain DB430 Δ*lipA* was included in these analyses as a reference. Diamide was added to a final concentration of 2 mM when the $OD_{600}$ had reached approximately 1.0 as was previously done by Leichert et al. (21). Samples were then harvested at 30 min, 1 h, and 4 h after diamide addition. A significant growth arrest after diamide addition could be observed for parental strain and Δ*ribT* deletion strain ($P < 0.01$) (Fig. 5). The Δ*pfkA* deletion strain, however, showed no significant growth arrest at 0.5 h after diamide addition ($P < 0.1$). It could be seen, however, that the variance of the $OD_{600}$ values for the control group of this was also larger at this time point. It appeared that the growth arrest of the Δ*pfkA* mutant was less severe, as an increase in $OD_{600}$ was already seen within 30 min of cultivation in the presence of 2 mM diamide (Fig. 5). A full recovery of growth was observed for this strain, as it reached the same $OD_{600}$ as the bacteria grown in the absence of diamide within 4 h after diamide addition.

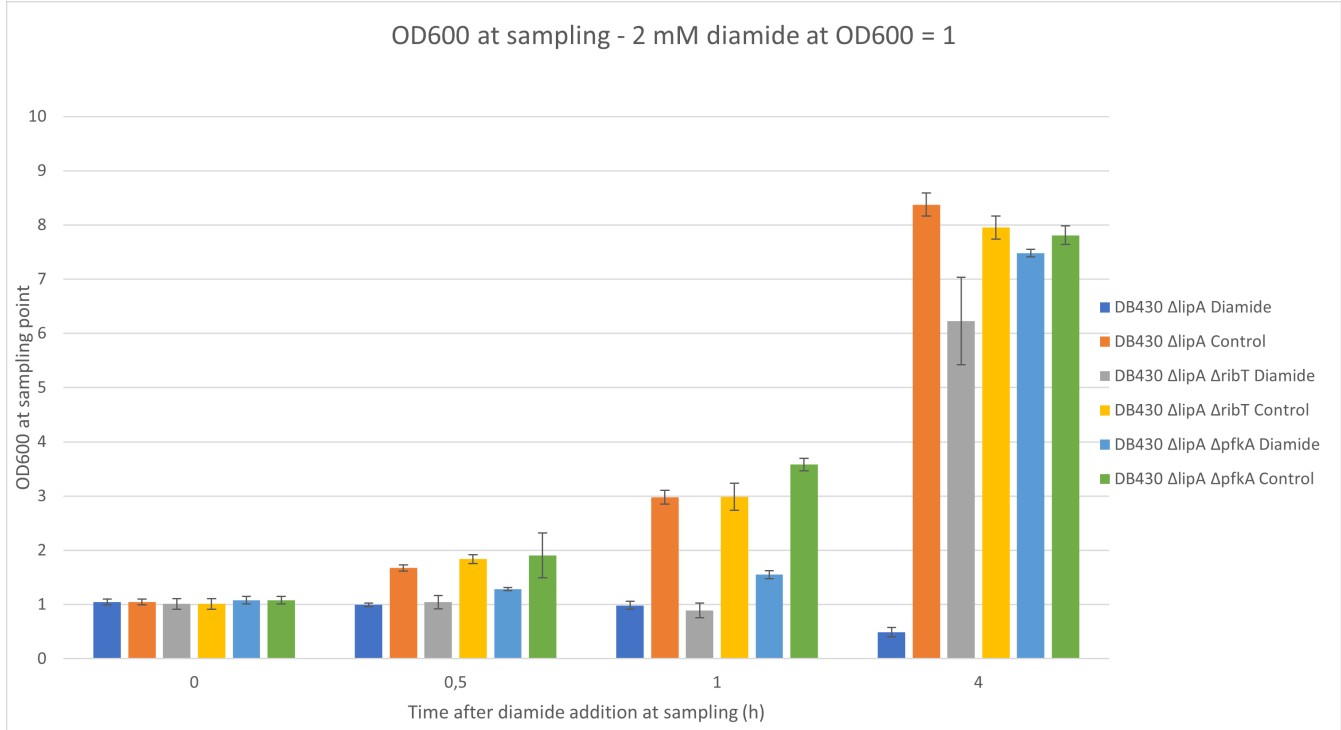

**FIG 5** Measurement of $OD_{600}$ of shake flask cultures of the parental strain (blue and orange) and *pfkA* or *ribT* deletion mutants. Optical densities were measured before the addition of diamide (0) and at 0.5, 1, and 4 h after the addition of 2 mM diamide. Bacterial cultures with no added diamide (control group) showed no growth inhibition. Bacterial cultures with 2 mM diamide showed growth phenotypes depending on the strain.

The Δ*ribT* strain was characterized by a prolonged growth delay, showing signs of lysis 1 h after diamide addition as indicated by a slight decrease in the $OD_{600}$. However, after 4 h, growth of the Δ*ribT* strain had been restored to normal with the bacteria reaching a similar $OD_{600}$ as the parental strain or the Δ*ribT* mutant grown in the absence of diamide. The parental strain DB430 Δ*lipA* did not recover from the diamide addition within the experimental timeframe and it showed decreasing optical densities at each sampling point up to 4 h after diamide addition.

### Proteome alterations in *pfkA* or *ribT* deletion mutants

Before focusing on the diamide response, the differences in strain behavior in the absence of diamide were given attention. The proteome of both deletion strains was compared to that of the parental strain, and the respective data are presented in Table S3. It was observed that the deletion of *pfkA* had a much larger impact on the proteome composition than the *ribT* deletion (Fig. 6 and 7). Comparing the Δ*pfkA* strain to the parental strain showed proteomic differences at all sampling points (Fig. 6). Sampling at 0.5 h, these differences were mostly related to proteins involved in metabolism (Table S3). However, after 1 h, an accumulation of stress-response-related proteins controlled by SigB, M, W, and X (Fig. S4) was evident in the *pfkA* deletion strain. In detail, mainly general stress proteins, but also oxidative stress proteins and heat shock proteins were found in higher amounts in the Δ*pfkA* strain. This included AhpC and KatA, proteins stemming from the peroxide regulon, which have already been shown to accumulate under diamide conditions (21). At 4 h, no difference was detectable in the abundance

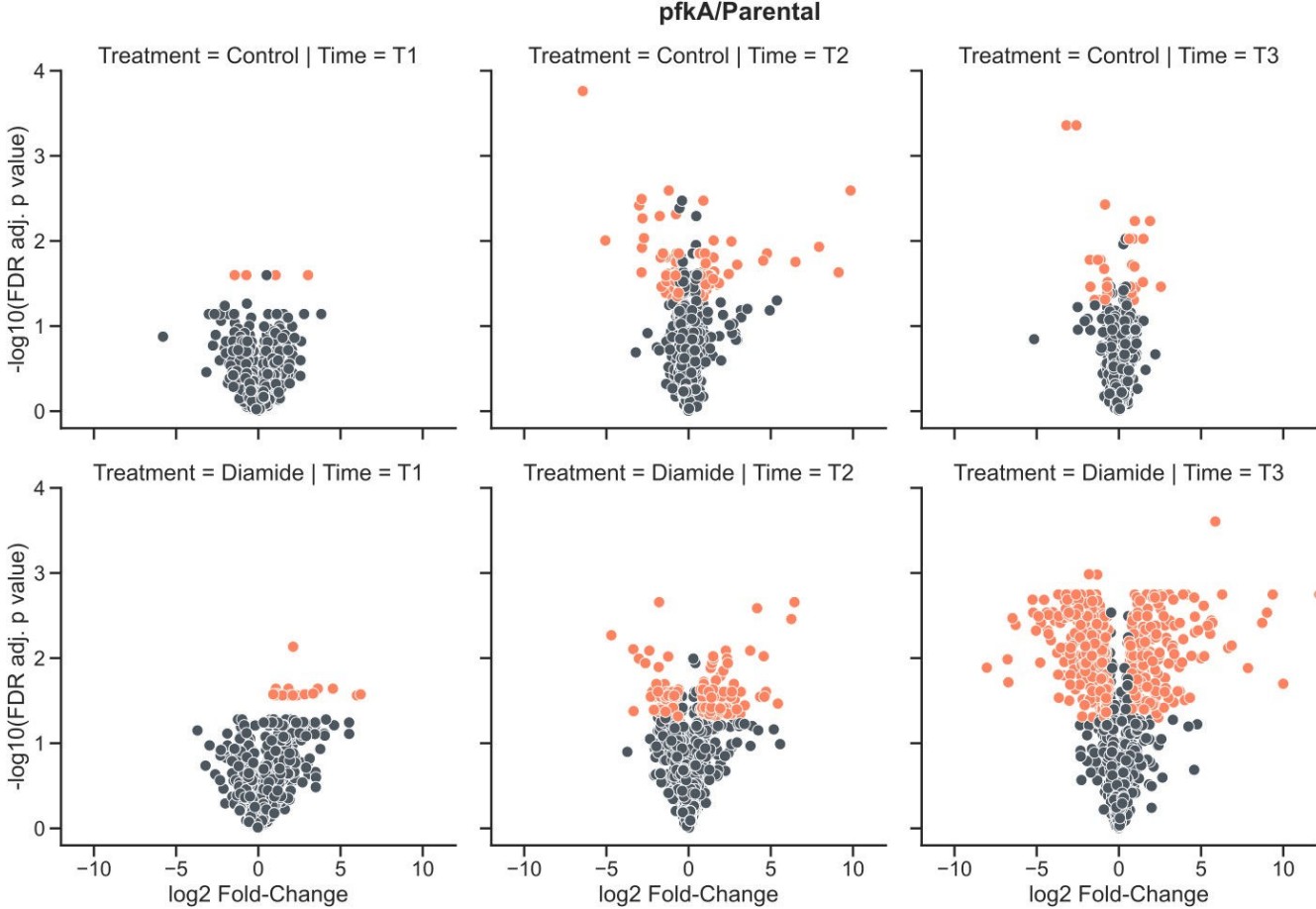

**FIG 6** Volcano plots comparing the abundance of proteins of the *pfkA* deletion strain and the parental strain in the presence of 2 mM diamide added after $OD_{600}$ reached 1.0. The upper and lower panels show differences in protein abundance at the different sampling points (0.5, 1, and 4 h) in the absence or presence of diamide, respectively. Proteins whose abundance was significantly changed are indicated in orange.

of stress-related proteins between the *pfkA* deletion strain and the parental strain. This indicated that either the deletion strain was no longer exhibiting signs of stress at this point, or rather both strains were now dealing with stress. The latter appeared more likely due to the nutrient depletion at the stationary phase activating the SigB stress response.

The comparison of the proteome of the Δ*ribT* strain to that of the parental strain revealed no significant differences when sampling after 0.5 or 1 h (Fig. 7). Only at 4 h, few differentially abundant proteins could be identified. However, these were mainly products encoded by the *rib* operon (RibBA, D, H), which had accumulated in the *ribT* deletion strain. In addition, the water deficit stress protein GsiB was accumulated in the *ribT* deletion strain, and a protein of unknown function, YugP, was depleted (Table S3).

## Redox state analysis

To determine the redox state of cysteine-containing proteins upon the addition of diamide, a differential cysteine labeling experiment was performed using culture samples harvested 0.5 h after imposing the diamide stress. To this end, natively reduced cysteines were initially blocked with standard iodoacetamide and, subsequently reversibly oxidized cysteines were reduced with TCEP and then labeled with heavy iodoacetamide-$^{13}C_2$, 2-$d_2$ (31). The percentage-wise rate of cysteine oxidation was determined for each protein in each sample by calculating the ratio of peptide ion intensities with oxidized cysteines compared to the total intensity of a peptide of a given protein. The calculated ratios were binned, which typically results in a distribution where higher rates of thiol oxidation are increasingly less abundant. Such a distribution was

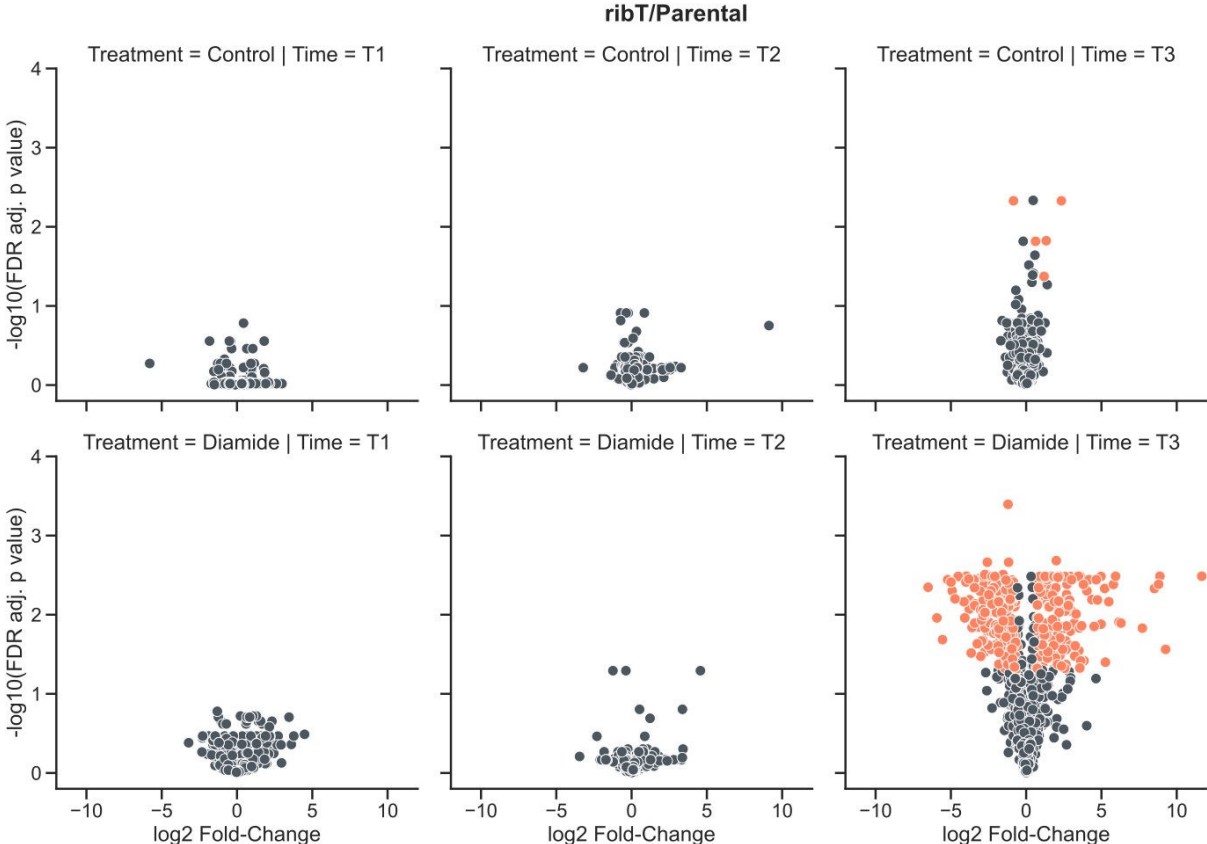

FIG 7 Volcano plots comparing the *ribT* deletion strain with the parental strain in the presence of 2 mM diamide added after OD$_{600}$ reached 1.0. At 4 h for the control group comparison (top right), the majority of proteins whose abundance significantly changed (orange) were those encoded by the *rib* operon. Overall, little difference was detectable in the proteomes of the two strains except for the diamide-treated group at 4 h after diamide addition, where the *ribT* deletion strain had resumed growth, while growth of the parental strain was still halted.

observed for the parental strain under control conditions (without diamide), where the largest group of proteins was oxidized for less than 5% (Fig. 8).

This is in sharp contrast to the results obtained for the *pfkA* deletion strain under control conditions, where the largest group (around 30%) of peptides were oxidized to an extent of 10%–20%. Under the same condition, the *ribT* deletion strain also showed a small shift toward higher oxidation (5%–10%). However, this difference was rather small, and the overall distribution of the oxidation bins was similar to those determined for the parental strain. In the presence of diamide, the Δ*pfkA* strain showed a distribution of cysteine oxidation bins similar to those of the parental strain, whereas the Δ*ribT* strain revealed a curve with a slightly less steep decline with more peptides in a higher oxidation state.

### Diamide stress response

From the proteomics analyses, it was evident that the proteomic response to the challenge with diamide took place between the 0.5- and 1-h sampling points (Fig. 9). This could be inferred from the relatively small amount of significantly changed protein abundances between the control group and the treated group 0.5 h after exposure to diamide, and the significant increase after 1 h exposure to diamide, which was seen in all three strains (Table S3). The respective protein abundance data are presented in Table S3.

At 1 h after the introduction of diamide, accumulation of proteins assigned to the following functional categories could be detected for all three strains (parental strain, Δ*pfkA*, and Δ*ribT*): oxidative stress response, heat shock proteins, chaperones, ribosomal proteins, and proteins involved in proteolysis.

When the respective oxidative stress responses from each strain compared to the control condition were analyzed, the *pfkA* deletion strain showed a high number of significant differences (Fig. 6; Table S3). In particular, the *pfkA* deletion strain showed the highest number of oxidative stress proteins (21 stress proteins) that were significantly more abundant in the presence of diamide compared to the control without diamide. This is in contrast to the parental strain, which presented nine more abundant stress proteins, and the *ribT* deletion strain, which presented 11 more abundant stress proteins upon challenge with diamide.

Notably, the BrxB protein associated with the remediation of S-thiolations (35) was more abundant (log2FC 1.11, adj. *P* < 0.05) in the diamide-treated group compared to the control group of the *pfkA* deletion strain after 1 h (Table S3). At the same time, in the

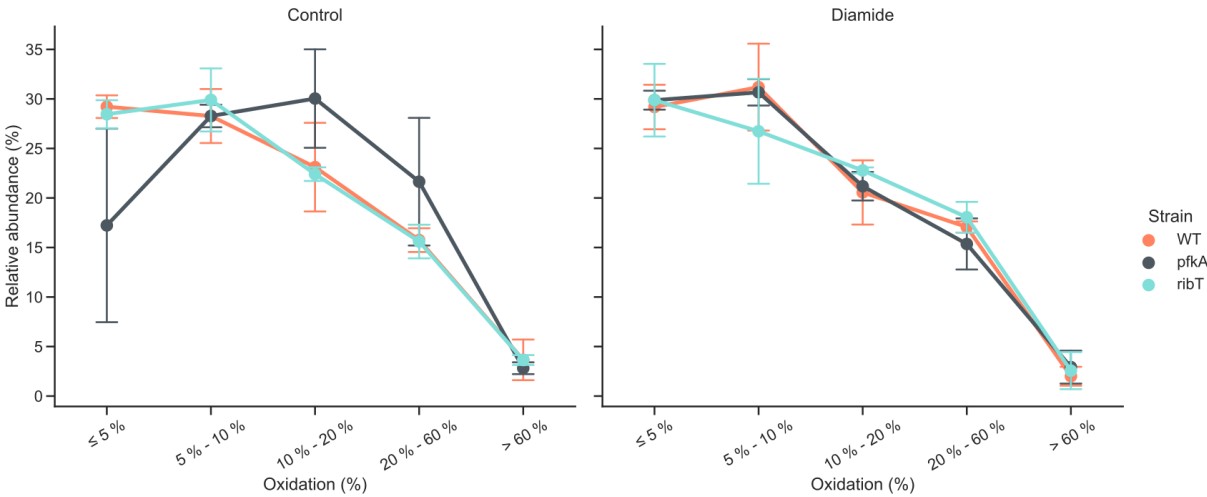

**FIG 8** Oxidation rates of peptidyl cysteine residues based on differential cysteine labeling measured for control (left) and diamide-treated groups (right) as determined 0.5 h after the introduction of 2 mM diamide or fresh medium, respectively. The data points represent the relative abundance of peptides with oxidized cysteine residues per "bin" showing the degree to which their cysteine residues were oxidized. The oxidation rates are shown for the parental strain (WT) as well as the Δ*pfkA* and Δ*ribT* mutants.

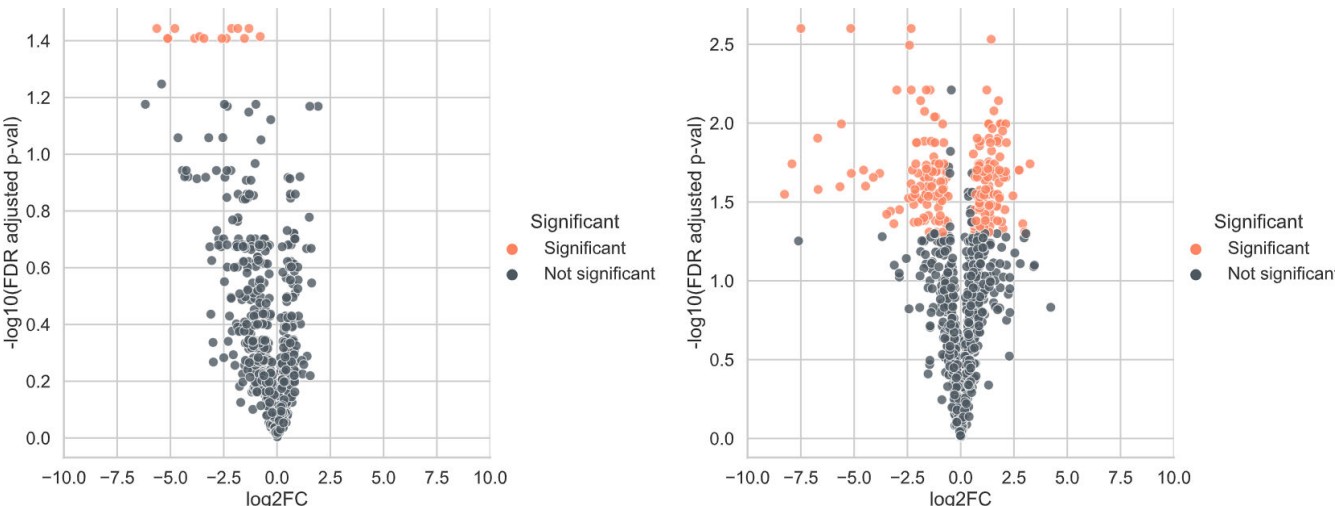

**FIG 9** Volcano plots of the parental strain showing relatively few significant differently abundant proteins (orange) after treatment with diamide after 0.5 h (left) compared to after 1 h (right). The two deletion strains showed a similar response (Fig. S3).

parental strain, BrxB was less abundant (log2FC −0.84, adj. *P* < 0.05) in the diamide group, while in the *ribT* deletion strain, no significant difference could be seen. After 4 h, when the *ribT* deletion strain had recovered, BrxB could now also be found as significantly more abundant (log2FC 0.68, adj. *P* < 0.05) in the diamide-treated group. The presence of this protein could be related to how the Δ*pfkA* strain and subsequently the *ribT* deletion strain were able to overcome the diamide inhibition (35).

Furthermore, the OhrA and OhrB proteins, which are commonly associated with oxidative stress (36, 37), were only found in diamide-treated samples for all three strains (Table S3). The protein OhrB was significantly more abundant in the *pfkA* deletion strain compared directly to the parental strain (log2FC 3.11, adj. *P* < 0.05). The *ohrA* gene is selectively expressed through a repression mechanism with OhrR (37). OhrB, however, is SigB-regulated (36), thus hinting at a higher general stress response in the *pfkA* deletion strain compared to the parental strain.

Interestingly, a strong accumulation (log2FC 2.56, adj. *P* < 0.01) of YqjM (NamA) was observed in the *pfkA* deletion strain after diamide treatment (Table S3). YqjM, which is a homolog of the "Old Yellow Enzyme" discovered in bottom brewer's yeast, has been found associated with the oxidative stress response and evidence exists that it also functions in the detoxification of xenobiotics (38). In addition, the paralogous protein YqiG, of which the function is not yet understood, was also more abundant in all three strains treated with diamide (Table S3). While its function in *B. subtilis* is still unclear, its ortholog in *Staphylococcus aureus* has been linked to remediating oxidative damage as caused by diamide (39). Notably, YqiG was also more abundant in the *ribT* deletion strain 4 h after diamide addition, when cell growth had recovered from diamide inhibition.

After 1 h, the *ribT* deletion strain showed few proteomic differences in its response to diamide when compared to the parental strain (Fig. 7; Table S3). Only after 4 h, when the *ribT* deletion strain had resumed growth, while the growth of the parental strain was still impaired, significant differences in protein abundance were seen. These differences were to a large extent growth phase-related, as exemplified by proteins related to chemotaxis, biofilm formation, and sporulation. Nevertheless, the higher abundance of BrxB and YqiG in the *ribT* deletion strain compared to the parental strain was similar to the proteomic adaptations detected in the *pfkA* deletion strain.

Aside from the accumulation of stress-related proteins, other differences in the proteomes of the parental strain and the *ribT* and *pfkA* deletion strains were also detectable upon diamide treatment (Table S3). For one, a strong accumulation could be observed for the ribosomal proteins in the parental strain and *ribT* deletion strain 1 h after exposure to diamide (Fig. S5 and S7). This accumulation was far less noticeable

in the *pfkA* deletion strain (Fig. S6). Furthermore, ribosomal proteins also accumulated when sampling at 4 h in the diamide-treated group of the parental strain compared to the non-treated group (average log2FC 3.5, adj. *P* < 0.05), while the abundance of ribosomal proteins did not significantly increase in the *ribT* or *pfkA* deletion strains that had recovered from diamide stress. This was unexpected as previous work by Leichert et al. described these proteins to be repressed under disulfide stress conditions (21). However, in line with the findings of this paper, an upregulation of proteins involved in cysteine biosynthesis was also detected in response to diamide stress in all three strains (Table S3).

## DISCUSSION

In this study, we present a novel and efficient method to screen a large library of transposon mutants in the context of disulfide stress imposed by diamide. Several mutant phenotypes with improved tolerance to the presence of diamide and, possibly, disulfide stress in general have been isolated. One of the strengths of this method is the direct identification of candidate mutants with improved disulfide stress tolerance due to the selective screening on diamide agar plates. This step greatly reduces the number of potential candidate mutants to be considered by selecting directly for the desired phenotype.

The insertion loci of the transposon in the selected mutants with improved disulfide stress tolerance were identified with an acceptable success rate of about 50%. Two-step gene walking methods are reliant on a certain amount of chance, as a suitable binding site for the degenerate primer must be found in proximity and the right orientation to the transposon insertion site. Further optimization could possibly improve on these results. The applied transposon mutagenesis method also created disruptions outside of coding regions of the *B. subtilis* genome and allowed the selection of desirable phenotypes without prior knowledge of function. While it was not an objective of our present study to analyze transposon insertions in non-coding regions, the method thus holds promise in relation to identifying the roles of these regions in regard to diamide resistance and disulfide stress adaptation. A recent study by Chai and colleagues also reported a link between *pchR* deletion and oxidative stress management (40). As *pchR* was also identified as a candidate gene in this study, this observation is further strengthening our claim for the high feasibility of our screening approach.

The disrupted genes identified during this study (Table 1) fulfilled a variety of functions. As was seen with the *cysE* disruption, these disruptions do not necessarily stem from the inactivation of the gene itself but could be due to a truncated gene being created, or even polar effects down the line. The identified phenotypes were validated independently for five out of six selected candidate mutants through a seamless gene deletion strategy. Excluding the possibility of off-target effects, the increased diamide tolerance was also evident in these deletion strains, thereby confirming the effectiveness of our diamide-based screening approach.

Reversible protein S-thiolations serve as a mechanism to protect active site cysteine residues from irreversible conversions to sulfonic acids or the formation of disruptive inter- and intramolecular disulfide bonds (23, 24). It has been shown that the response to diamide in *B. subtilis* and *S. aureus* causes mainly reversible protein S-thiolations with LMW thiols (23). These thiolations also influence cellular processes. A down-regulation of household genes by 80% due to the S-thiolation-regulated stringent response triggered by diamide was reported in the study by Leichert and colleagues (21). This would also contribute to the growth arrest that occurred upon introduction of the compound. However, when looking at the proteomic data generated in this study, this could not be confirmed as a higher abundance of ribosomal proteins, regulated by the stringent response, was witnessed under diamide duress. This strong accumulation was not detectable to a significant degree in bacteria that had recovered from diamide stress and could, thus, be related to the diamide treatment.

Interestingly, disrupting *bshC*, a gene involved in the biosynthesis of bacillithiol (41), was found to be beneficial to diamide resistance in this study (Table 2; Fig. 3). Bacillithiol was studied by Gaballa and colleagues, who were able to show no negative effects on the oxidative stress response and only a modest increase in the sensitivity to diamide and methyl glyoxal for *bshA* null mutants, which are devoid of bacillithiol (41). The overlap in function between bacillithiol and cysteine is, therefore, thought to preclude severely negative effects in the presence of diamide (41). BshC catalyzes the final step in the biosynthesis of bacillithiol, transferring a cysteine to the intermediary, and null mutants for *bshC* are also devoid of bacillithiol (41). However, it is conceivable that preventing this reaction allowed to free up more cysteines to interact with diamide, thereby clearing the diamide-imposed inhibition faster.

Focusing on the most promising candidate mutants that we identified in our screening for diamide tolerance, the *ribT* and *pfkA* deletion strains were further investigated to elucidate the mechanism behind their diamide-tolerant phenotype. Based on the proteomics data generated toward this end, significant differences could be detected both in the proteome and the protein redox state of the *pfkA* deletion mutant and the parental strain. This was actually evident both in the absence or presence of diamide. Opposed to the *ribT* deletion, deletion of the *pfkA* gene had a more global effect on the cell and the resulting phenotype could, therefore, not have been directly related to the function of phosphofructokinase (PFK), which is the *pfkA* gene product. Interestingly, the increased cysteine residue oxidation observed in the *pfkA* deletion strain in the absence of diamide seems to suggest that the strain is perceiving oxidative stress conditions regardless of the presence of diamide. This idea is actually verified by the proteomics data, where an accumulation of proteins involved in the SigB stress response and the oxidative stress response was detected. This implies that the *pfkA* deletion strain was already expressing stress response proteins in the absence of diamide. It can be argued that such a state of enhanced readiness allows the strain to overcome the effects of diamide more easily, which is reminiscent of how cells primed with a low dose of $H_2O_2$ are later able to overcome an otherwise lethal dose of this oxidative reagent (10).

Based on previous publications, it may be hypothesized that the diamide-tolerant phenotypes of the *pfkA* and *ribT* deletion mutants could be due to their relation to the cellular NADPH pool. Many bacterial species belonging to the *Firmicutes,* including *B. subtilis,* utilize LMW thiols, such as the coenzyme A-SH, cysteine, and bacillithiol as major thiol-redox buffers (23, 42). In *B. subtilis*, the predominant LMW thiol is bacillithiol. During the stationary phase, bacillithiol is present in a concentration about 17-fold higher than that of cysteine which is, in fact, present at a relatively low concentration (43). Importantly, both these LMW thiols help the cell to regulate the cellular redox state and perform thiol protection during conditions of oxidative stress, such as the stress conditions imposed by diamide (11, 23, 42). Furthermore, reversing the S-thiolations is NADPH intensive (44). This is due to the need for NADPH-dependent thioredoxins to perform the reduction of the formed intracellular disulfides (21, 45, 46). In *B. subtilis*, this reversal is performed by the thioredoxin-like enzymes BrxA (YphP, YpdA), BrxB (YqiW), and BrxC (YtxJ) (35, 47, 48). During this study, it was seen that strains recovering from diamide inhibition presented a higher abundance of the BrxB protein, supporting its importance for diamide resistance.

In the case of the *ribT* deletion strain, the proteomic data indicated only relatively moderate effects of the deletion. The mechanism behind the diamide tolerance is thus likely resulting directly from the protein function of RibT. The GN5-related acetyltransferase RibT is involved in the synthesis of riboflavin (49). This protein is not essential for riboflavin synthesis but, when absent, intracellular riboflavin levels are decreased (49, 50). Cytoplasmic riboflavin is converted rapidly and almost entirely into flavonoid mononucleotides (51). It has been shown that the presence of these flavonoid mononucleotides derived from riboflavin will increase the cellular rate of NADPH oxidation (52). Therefore,

we postulate that reducing the intracellular levels of riboflavin could make more NADPH available for counteracting the detrimental effects of diamide.

A relationship with the cellular NADPH pool could also be applicable for the *pfkA* deletion strain judged by prior work of Chavarría and colleagues (44). In *Pseudomonas putida*, the addition of the *Escherichia coli pfkA* gene, coding for 6-phosphofructokinase, to open the Emden-Meyerhoff pathway led to an increased susceptibility to diamide. The reason for this was proposed to be the increased demand for NADPH due to the activity of PFK (44). This idea would be in line with the phenotype observed in our present study upon mutation of the *B. subtilis pfkA* gene, for which six transposon insertion mutants were independently selected. However, a major difference between our study and previous studies by others concerns the use of LB medium for screening purposes. This complex medium is widely used, but it contains only small amounts of fermentable sugars that are rapidly depleted (53). Since PFK is a part of the glycolysis pathway, which converts glucose to pyruvate, it is possible that this is not the role linked to the beneficial effects of the *pfkA* deletion on diamide tolerance as documented in our present study.

Intriguingly, PfkA was previously shown to moonlight in a secondary role in the RNA degradosome (54). The same is true for another gene disruption identified in this study, where the transposon was inserted in the *rny* gene for RNase Y, which also interacts with PfkA (54, 55). RNase Y has been suggested to be the functional equivalent of RNase E in *E. coli* (56). The reduced activity of RNases would result in increased persistence of mRNA, leading to a higher rate of protein translation. A similar and rather sophisticated regulatory mechanism was described in *E. coli* for the oxidative stress protein TrxB, of which the NP4H-capped mRNA persists longer in the presence of diamide due to cross-linking of the cysteines of the de-capping enzyme ApaH (57). Likewise, reducing the RNA degradosome in *B. subtilis* DB430 Δ*lipA* would be an analogous mechanism to counteract the detrimental effects of diamide. Overall, the exact mechanism by which the *pfkA* deletion strain achieved its diamide-tolerant phenotype is hard to pinpoint. It could relate to a reduced demand on the NADPH pool, a reduced availability or activity of the RNA degradosome, or an increased expression of stress management proteins. Most likely, the diamide tolerance is due to a combination of these effects of the *pfkA* gene deletion.

## Conclusion

Altogether, we conclude that the screening method presented here is effective in identifying disulfide stress-tolerant phenotypes in (transposon) mutant libraries. The encountered phenotypes could be traced back to different genomic disruptions and the mechanisms behind these phenotypes were also distinct. The identified mutations could potentially be implemented in future strain engineering strategies to improve existing production strains that are sensitive to oxidative stress. Our method also holds promise to incorporate non-coding regions into these strategies. Finally, the method could also be complemented using a transposon carrying a strong inducible promoter to induce overexpression of downstream genes upon transposition.

## ACKNOWLEDGMENTS

We thank Dr. Anne de Jong for determining the precise number of TA motifs in the *B. subtilis* genome. We would like to express our gratitude to Lilly Franzmeyer for her technical assistance in sample preparation for proteomics analyses.

This work was supported by the European Commission as part of the Horizon 2020 EU-funded project 813979 Secreters.

J.W., R.S., and M.S. are employees of AB Enzymes GmbH. All other authors agree there are no competing interests.

## AUTHOR AFFILIATIONS

[1]AB Enzymes GmbH, Darmstadt, Germany

²Department of Microbial Proteomics, University of Greifswald, Greifswald, Germany
³Department of Medical Microbiology, University of Groningen, University Medical Center Groningen, Groningen, the Netherlands

## AUTHOR ORCIDs

Sandra Maaß  http://orcid.org/0000-0002-6573-1088
Jan Maarten van Dijl  https://orcid.org/0000-0002-5688-8438
Michael Seefried  http://orcid.org/0009-0002-8843-6521

## FUNDING

| Funder | Grant(s) | Author(s) |
|---|---|---|
| EC \| H2020 \| PRIORITY 'Excellent science' \| H2020 Marie Skłodowska-Curie Actions (MSCA) | 813979 | Jonathan Walgraeve |
| | | Borja Ferrero-Bordera |

## AUTHOR CONTRIBUTIONS

Jonathan Walgraeve, Conceptualization, Data curation, Formal analysis, Investigation, Methodology, Visualization, Writing – original draft, Writing – review and editing | Borja Ferrero-Bordera, Data curation, Formal analysis, Investigation, Visualization, Writing – review and editing | Sandra Maaß, Data curation, Formal analysis, Supervision, Writing – review and editing | Dörte Becher, Funding acquisition, Resources, Writing – review and editing | Ruth Schwerdtfeger, Funding acquisition, Project administration, Resources, Writing – review and editing | Jan Maarten van Dijl, Funding acquisition, Supervision, Writing – review and editing | Michael Seefried, Conceptualization, Funding acquisition, Methodology, Project administration, Resources, Supervision, Writing – original draft, Writing – review and editing

## DATA AVAILABILITY

The proteome data set has been deposited to the ProteomeXchange Consortium via the PRIDE partner repository (58) with the dataset identifier PXD041512.

## ADDITIONAL FILES

The following material is available online.

### Supplemental Material

**Fig. S1 (Spectrum01608-23-s0001.pdf).** Establishing the suitable diamide inhibition concentration for the parental strain DB430 Δ*lipA* for the microtiter plate kinetic growth assay.
**Fig. S2 (Spectrum01608-23-s0002.pdf).** Variation in growth inhibition times indicated by the time to reach an OD600 of 0.5 for the parental strain at different diamide concentrations.
**Fig. S3 (Spectrum01608-23-s0003.pdf).** Volcano plots comparing the abundance of proteins of each investigated strain at the different sampling points (0.5 h, 1 h, and 4 h) in the absence or presence of diamide, respectively.
**Fig. S4 (Spectrum01608-23-s0004.jpg).** Regulon sorted Voronoi tree map comparing protein abundance in the *pfkA* deletion strain and parental strain in the absence of diamide at 1 h after the addition of diamide.
**Fig. S5 (Spectrum01608-23-s0005.jpg).** Subtiwiki classification sorted Voronoi tree map of the parental strain DB430 Δ*lipA* at 1 h after diamide addition.
**Fig. S6 (Spectrum01608-23-s0006.jpg).** Subtiwiki classification sorted Voronoi tree map of the *pfkA* deletion strain at 1 h after diamide addition.

**Fig. S7 (Spectrum01608-23-s0007.jpg).** Subtiwiki classification sorted Voronoi tree map of the *ribT* deletion strain at 1 h after diamide addition.

**Tables S1 and S2 (Spectrum01608-23-s0008.docx).** Primer list and summary of transposition events and CFU counts to estimate the total number of transposon mutants screened during the study.

**Table S3 (Spectrum01608-23-s0009.xlsx).** Proteomics data.

## Open Peer Review

**PEER REVIEW HISTORY (review-history.pdf).** An accounting of the reviewer comments and feedback.

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
