## [Reviewer comments · Microbiology Spectrum]

Microbiology Spectrum

Diamide-based screening method for isolation of improved oxidative stress tolerance phenotypes in *Bacillus* mutant libraries.

Jonathan Walgraeve, Borja Ferrero-Bordera, Sandra Maaß, Dörte Becher, Ruth Schwerdtfeger, Jan Maarten van Dijl, and Michael Seefried

Corresponding Author(s): Michael Seefried, AB Enzymes GmbH

Review Timeline:

Submission Date:	April 17, 2023
Editorial Decision:	June 26, 2023
Revision Received:	August 29, 2023
Accepted:	August 30, 2023

Editor: Sacha Pidot

Reviewer(s): The reviewers have opted to remain anonymous.

Transaction Report:

DOI: <https://doi.org/10.1128/spectrum.01608-23>

June 26, 2023

Dr. Michael Seefried
AB Enzymes GmbH
Feldbergstrasse 78
Darmstadt 64293
Germany

Re: Spectrum01608-23 (Diamide-based screening method for isolation of improved oxidative stress tolerance phenotypes in *Bacillus* mutant libraries.)

Dear Dr. Michael Seefried:

Please find below (and attached) the reviewers comments for your manuscript. Overall, the reviewers thought the article merited publication, but that the text requires some modifications before it can be accepted for publication in Microbiology Spectrum. Please note the reviewers did not recommend any further experimental work, but rather rephrasing and rewriting of certain sections.

Link Not Available

Sincerely,

Sacha Pidot

Journals Department
Reviewer comments:

Reviewer #1 (Comments for the Author):

This paper elucidate two aspect of interest in *Bacillus subtilis* biotechnology: the role and pleiotropic effects of PfkA and RibT on diamide induced stress, and theTn mutagenesis as a strategy to find new genes involved in disulfide stress, without previous knowledge of sequences or functions, and to plan strain improvements by seamless targeted mutagenesis. The role of these two functions in future could be confirmed by gain of function approaches.

The results suggest a role concerning of the NADPH pool and of the RNA degradosome in the Bacillus response to oxidative stress and open the possibility of strain improvement for industrial production.

Reviewer #2 (Comments for the Author):

Oxidative stress tolerance

1. Fig.1 why do 0 and 100uM look different?
2. P.10, line9- "effective"- does this mean time of diamide stability? Line 12- undiluted-what does this mean as to cell concentration?
3. P.11, line 11- what is the Shortest delay?
4. Fig2- is the strain with the lowest growth delay ribT? Might help to indicate in legend.
5. Fig. 3- how significant are the differences at 2mM?
6. Fig.4-describe shaded area at 6mM. Is this the ribT mutant? All the other plots show no differences not even 4mM no follow up on bshC so why include here? Explain.
7. P. 12- 6 lines up- why OD 1.0?
8. Fig. 5 and p.12- 3 lines up; 0.5h differences not evident.
9. P.13, line 4- where is the data? Paragraph 4- is there data for YugP and GsiB statement?
10. Figs 6&7- indicate diamide concentration. Where is data to show metabolism involvement as stated on p.13?
11. P. 14, top plus Fig. 8- no error bars in control plot to show real differences. What does slightly flatter curve mean?
12. P. 14-paragraph 3- how many peptides for each protein were used to identify each class of proteins? Might include in Table 1.
13. P.14-15 "Notably" all these statements are not supported by data so why include?
14. P.16 7 lines up- what is the significance of the ribosomal protein changes? Is this reflected in rRNA as well?
15. P.19 line 4- what does "variable" mean and why?
16. Introduction- explain "more robust" and cite evidence.
17. P.8 details in paragraphs 2&3 can be shortened.
18. Do other oxidizing agents cause similar changes in designated genes or are they unique to diamide? Not clear what the value of resistant strains is and is there data for this? Would be valuable information to include.

Staff Comments:

Preparing Revision Guidelines

- Point-by-point responses to the issues raised by the reviewers in a file named "Response to Reviewers," NOT IN YOUR COVER LETTER.
- Upload a compare copy of the manuscript (without figures) as a "Marked-Up Manuscript" file.

- Each figure must be uploaded as a separate file, and any multipanel figures must be assembled into one file.
- Manuscript: A .DOC version of the revised manuscript
- Figures: Editable, high-resolution, individual figure files are required at revision, TIFF or EPS files are preferred

Please return the manuscript within 60 days; if you cannot complete the modification within this time period, please contact me. If you do not wish to modify the manuscript and prefer to submit it to another journal, please notify me of your decision immediately so that the manuscript may be formally withdrawn from consideration by Microbiology Spectrum.

This paper elucidate two aspect of interest in *Bacillus subtilis* biotechnology: the role and pleiotropic effects of PfkA and RibT on diamide induced stress, and the Tn mutagenesis as a strategy to find new genes involved in disulfide stress, without previous knowledge of sequences or functions, and to plan strain improvements by seamless targeted mutagenesis. The role of these two functions in future could be confirmed by gain of function approaches.

The results suggest a role concerning of the NADPH pool and of the RNA degradosome in the *Bacillus* response to oxidative stress and open the possibility of strain improvement for industrial production.

The paper needs some revisions before publication, here listed:

1, Major revisions

Authors should:

- Pg.4 Add a review reference on examples of Tn mutagenesis to obtain different independent bacterial mutants;
- Pg.10 Comment about the possibility that also a prolonged treatment with Diamide may cause "stress directed mutagenesis" and produce undesirable mutants. See Sung and Yasbin, JOURNAL OF BACTERIOLOGY, Oct. 2002, 184(20), 5641–5653;
- Pg.10 and Supplementary Table 2: the amount of viable screened CFU seems incoherent with the data in Supplementary Table 2, possibly TMA1 and TMA4 viable amounts screened are +04?
- Pg.10 5.85×10^5 is the number of estimated TA sites, Himar1 targets, in *B. subtilis* genome? If yes specify;
- Pg.13 and Supplementary Fig.4: SigV regulon don't show, in presence of *pfkA* deletion, higher abundance proteins (red hexagon); moreover where are shown SigX and SigY regulons mentioned in Pg.13?
- Pg. 13 "the percentage-wise rate of cysteine oxidation was determined for each protein.." or "for each protein sample"?
- Pg. 13-14 The sentence "The calculated ratios were binned, which is expected to result in a downward sloping curve. Indeed, such a curve was observed for the parental strain under control conditions (without diamide), where the largest group of proteins was oxidized..." is unclear;
- Pg.14 "At 1 h after introduction of diamide, accumulation of proteins assigned to the following functional categories could be detected for all three strains (parental, $\Delta pfkA$, $\Delta ribT$)..", but in Fig3 Supplementary materials also in the parental strain after 1 hr of diamide exposure the amount of proteins is significantly changed;
- Pg. 16 Why Authors say that Tn mutagenesis was not part of the study. It was not the purpose of the described work, but was an essential part and a good methodological choice;
- Pg.17 In Discussion more comments are needed about the results observed concerning BrxB protein (see Pg.14).
- Pgs. 21, 22, 23 and others, each References citation need careful revisions from the editing point of view: see cit 15, names instead of family names for some Authors or for example cit 42 *Saccharomyces* that should be in capital. Everywhere use italics for the species names.

2. Minor revisions or editing problems:

- Pg.6 in Material and Methods the medium LB according to literature is Luria and Bertani Medium and not Lysogeny medium, see also your ref. 50;
- Pg.6 and at pg.7, in Material and Methods, the paragraphs "An overnight culture in All plates were incubated at 37C" is repeated twice at pg. 6 under the title "Kinetic diamide growth assay" where it would be off topic and probably an unnecessary section;

- Pg.7 in Materials and Methods, specify the experimental provenance of the “deep well plates stored at ..”;
- Pg. 8 in Materials and Methods, in differential cysteine labeling, the switch of labeling order of the third sample was done as a control?
- Pg10 in Library generation and coverage, refer to Supplementary Table 2 not 1;

Manuscript No. Spectrum01608-23

'Diamide-based screening method for isolation of improved oxidative stress tolerance phenotypes in *Bacillus* mutant libraries' (Walgraeve et al.)

Authors' response to the comments of the Reviewers

Please note that the comments by the two Reviewers were divided up and marked in black. Our responses are marked in blue. Please note also that all references to page and line numbers in our responses below relate to the 'Track Changes' version of our manuscript in which all revisions have been marked with the Track Changes tool of Word.

Author's response to the comments of Reviewer #1:

This paper elucidate two aspect of interest in *Bacillus subtilis* biotechnology: the role and pleiotropic effects of PfkA and RibT on diamide induced stress, and the Tn mutagenesis as a strategy to find new genes involved in disulfide stress, without previous knowledge of sequences or functions, and to plan strain improvements by seamless targeted mutagenesis. The role of these two functions in future could be confirmed by gain of function approaches.

The results suggest a role concerning of the NADPH pool and of the RNA degradosome in the *Bacillus* response to oxidative stress and open the possibility of strain improvement for industrial production.

The paper needs some revisions before publication, here listed:

Response: We thank the Reviewer for the constructive comments and suggestions, which have helped us to improve our manuscript.

- Pg.4 Add a review reference on examples of Tn mutagenesis to obtain different independent bacterial mutants;

Response: As proposed by the Reviewer, we have included additional references on transposon mutagenesis in our revised manuscript (new Refs 18-20, page 4, line 81).

- Pg.10 Comment about the possibility that also a prolonged treatment with Diamide may cause "stress directed mutagenesis" and produce undesirable mutants. See Sung and Yasbin, JOURNAL OF BACTERIOLOGY, Oct. 2002, 184(20), 5641–5653;

Response: We appreciate the comment of the Reviewer. Accordingly, we have addressed the need of avoiding the accumulation of undesired spontaneous mutations that are not related to a transposon insertion on page 10 (lines 293-294).

- Pg.10 and Supplementary Table 2: the amount of viable screened CFU seems incoherent with the data in SupplementaryTable 2, possibly TMA1 and TMA4 viable amounts screened are +04?

Response: In response to the comment of the Reviewer, we have rephrased this paragraph to avoid misunderstandings by the readers of our manuscript. It now reads as follows: "Mutant libraries were generated by four cycles of transposon mutagenesis, plated on LB agar containing 10 µg/mL kanamycin and 200 µM diamide, and incubated at 37 °C. Colonies were picked at 16 h, 24 h, and 40 h after plating and transferred to fresh LB-kanamycin plates. Thus, a total of 312 candidate colonies of bacteria with potentially increased diamide tolerance was selected. In parallel to the selection of diamide tolerant transposon mutants, CFUs were counted on LB plates with kanamycin, but without

diamide, in order to estimate the total number of bacteria with transposon mutations (Supplementary table 2). These plates were incubated at 37 °C to avoid plasmid replication. The results showed that an average viable count of $1.4 \pm 0.7 \times 10^5$ CFU/mL was obtained after heat shock. With 100 μ l aliquots plated on 97 plates, this implied that a total of about $1.4 \pm 0.7 \times 10^6$ viable transposition events were screened for diamide tolerance" (page 10, lines 302-311).

- Pg.10 5.85×10^5 is the number of estimated TA sites, Himar1 targets, in *B. subtilis* genome? If yes specify;

Response: We originally estimated the theoretical number of TA sites available for transposon integration by the length (4.2 Mb) and GC content (43.5%) of the *B. subtilis* genome, and by assuming a random distribution of nucleotides. However, we have now determined that there are precisely 561325 TA sites in the *B. subtilis* 168 genome. This is now specified on page 10 (lines 312-314).

- Pg.13 and Supplementary Fig.4: SigV regulon don't show, in presence of *pfkA* deletion, higher abundance proteins (red hexagon); moreover where are shown SigX and SigY regulons mentioned in Pg.13?

Response: Indeed, we observed no induction of SigV-regulated proteins in the *pfkA* deletion strain at 1 hour after addition of diamide. SigX-regulated proteins are shown on the left lower half of the Voronoi treemap. Indeed, there are no SigY-regulated proteins included as these proteins were not identified in our proteome analysis. We have corrected this in the manuscript (page 13, line 446).

- Pg. 13 "the percentage-wise rate of cysteine oxidation was determined for each protein.." or "for each protein sample"?

Response: Indeed, the percentage-wise rate of cysteine oxidation was determined for each protein in each sample. We have now clarified this (page 13, lines 466-468).

- Pg. 13-14 The sentence "The calculated ratios were binned, which is expected to result in a downward sloping curve. Indeed, such a curve was observed for the parental strain under control conditions (without diamide), where the largest group of proteins was oxidized..." is unclear;

Response: We thank the Reviewer for pointing out that this sentence was unclear and have rephrased it as follows: "The calculated ratios were binned, which typically results in a distribution where higher rates of thiol oxidation are increasingly less abundant. Such a distribution was observed for the parental strain under control conditions (without diamide), where the largest group of proteins was oxidized for less than 5% (Figure 8)" (pages 13-14, lines 468-478).

- Pg.14 "At 1 h after introduction of diamide, accumulation of proteins assigned to the following functional categories could be detected for all three strains (parental, Δ pfkA, Δ ribT)..", but in Fig3 Supplementary materials also in the parental strain after 1 hr of diamide exposure the amount of proteins is significantly changed;

Response: The Reviewer is right that also the proteome of the parental strain showed significant changes at 1 hour after diamide addition as shown in Supplementary figure 3. In fact, we had indicated this with the word "parental" in parenthesis. We have now made this more clear by specifying "parental strain" (page 14, line 494). Furthermore, we have made this more explicit by rephrasing lines 488-491 (page 14), which now read as follows: "This could be inferred from the

relatively small amount of significantly changed protein abundances between the control group and the treated group 0.5 h after exposure to diamide, and the significant increase after 1 h exposure to diamide, which was seen in all three strains (Supplementary figure 3)". We furthermore refer to Supplementary table 3 for the respective protein abundance data (page 14, lines 491-492).

- Pg. 16 Why Authors say that Tn mutagenesis was not part of the study. It was not the purpose of the described work, but was an essential part and a good methodological choice;

Response: We appreciate the understanding of the Reviewer. Indeed, it was not an objective of our present study to analyse transposon insertions in non-coding regions and we have now stated this more explicitly (page 16, lines 568-570).

- Pg.17 In Discussion more comments are needed about the results observed concerning BrxB protein (see Pg.14).

Response: We thank the Reviewer for giving us the opportunity to expand the Discussion section with respect to the role of BrxB. Accordingly, we have added the following sentence on page 17 (lines 631-633): "During this study, it was seen that strains recovering from diamide inhibition presented a higher abundance of the BrxB protein, supporting its importance for diamide resistance".

- Pgs. 21, 22, 23 and others, each References citation need careful revisions from the editing point of view: see cit 15, names instead of family names for some Authors or for example cit 42 *Saccharomyces* that should be in capital. Everywhere use italics for the species names.

Response: We apologize for the typographic mistakes in the Reference list. We have now carefully re-read the list and corrected the mistakes.

2. Minor revisions or editing problems:

- Pg.6 in Material and Methods the medium LB according to literature is Luria and Bertani Medium and not Lysogeny medium, see also your ref. 50;

Response: We agree with the Reviewer that, in the literature, the abbreviation LB is often considered to refer to Luria and Bertani, as is exemplified by Ref 54 (i.e. the former Ref 50). However, in the original publication, Bertani defined this medium as Lysogeny Broth (LB) and for this reason we have defined LB as such.

- Pg.6 and at pg.7, in Material and Methods, the paragraphs "An overnight culture in All plates were incubated at 37C" is repeated twice at pg. 6 under the title "Kinetic diamide growth assay" where it would be off topic and probably an unnecessary section;

Response: We apologize for the mistake which must have occurred during the assembly of the final version of our manuscript. The duplication on page 7 (i.e. page 6 of the original manuscript) has been removed and replaced with the appropriate description of the kinetic diamide growth assay (page 6, lines 136-147).

- Pg.7 in Materials and Methods, specify the experimental provenance of the "deep well plates stored at ..";

Response: In response to this comment of the Reviewer, we have rephrased the sentences referring to the deep well plates stored at -80 °C. The sentence now reads as follows: "The deep well plates

containing the overnight cultures for the kinetic growth assay, were taken from the -80 °C storage and thawed" (page 7, lines 181-182).

- Pg. 8 in Materials and Methods, in differential cysteine labeling, the switch of labeling order of the third sample was done as a control?

Response: Indeed, the switch in the labeling order was done as a control for the labeling procedure, in order to exclude potentially marginal effects that could be caused by the different labels. This has now been specified on page 8 (lines 244-245).

- Pg10 in Library generation and coverage, refer to Supplementary Table 2 not 1;

Response: We thank the Reviewer for pointing out the mistake which has been corrected (page 10, line 308).

Author's response to the comments of Reviewer #2:

Oxidative stress tolerance

Response: We thank the Reviewer for the constructive comments and suggestions, which have helped us to improve our manuscript.

1. Fig.1 why do 0 and 100uM look different?

Response: In the absence of diamide treatment (0 μM), the plate is entirely overgrown by bacterial colonies, whereas defined colonies become visible in the presence of 100 μM diamide. We have added a caption in the legend to Figure 1 in order to clarify this effect (page 27).

2. P.10, line9- "effective"- does this mean time of diamide stability? Line 12- undiluted-what does this mean as to cell concentration?

Response: We have replaced the word "effective" with "practical" to point out that we are not referring to effective diamide treatment but practical and manageable experimental time. However, we also considered the possibility that extended incubation in the presence of diamide would also increase the chance of accumulating undesired spontaneous mutations that are not related to a transposon insertion. This has been clarified on page 10 (lines 291-294).

With "undiluted", we refer to the fact that the cell suspensions were heat-shocked after 5 h cultivation at 37°C to induce transposition (see page 7, lines 174-175). At this point, the OD600 was typically just over 3. Normally, plating of a bacterial suspension with this OD would result in confluent growth, but at a diamide concentration of 200 μM no dilution of the bacterial suspension was necessary to obtain individual colonies of diamide tolerant bacteria upon plating. We have clarified this by referring to "heat-shocked cell suspensions" (page 10, line 296).

3. P.11, line 11- what is the Shortest delay?

Response: The shortest growth delays in the presence of diamide for the selection of diamide tolerant mutants fell in a window of 1 to 15 h (see Figure 2). This has now been specified on page 11 (lines 350-351).

4. Fig2- is the strain with the lowest growth delay ribT? Might help to indicate in legend.

Response: Actually, the mutant with the shortest growth delay in this assay was a *pchR* mutant. We have now mentioned this in the legend of Figure 2 (page 27).

5. Fig. 3- how significant are the differences at 2mM?

Response: The significance of the results presented in Figure 3 was tested and the outcomes are presented on page 12 (lines 385-387).

6. Fig.4-describe shaded area at 6mM. Is this the *ribT* mutant? All the other plots show no differences not even 4mM no follow up on *bshC* so why include here? Explain.

Response: In the plot of Figure 4 that depicts the growth of the *ribT* mutant at 6 mM diamide, the area shaded in green marks the variation in the individual growth curves. This has now been specified in the legend of Figure 4 (page 27).

7. P. 12- 6 lines up- why OD 1.0?

Response: Diamide was added when the cultures reached an OD₆₀₀ of 1, based on the protocol of Leichert et al. To make this clear for the reader, we have now referred to the publication by Leichert et al. (Ref. 21) on page 12 (lines 417-419).

8. Fig. 5 and p.12- 3 lines up; 0.5h differences not evident.

Response: At 0.5 h after diamide addition, significant growth inhibition in the presence of diamide is evident for all tested strains as shown by t-tests. This information has now been included in our manuscript (page 12, lines 419-423).

9. P.13, line 4- where is the data? Paragraph 4- is there data for YugP and GsiB statement?

Response: The data is presented in Supplementary table 3, which has now been specified on page 13 (line 441 and line 460).

10. Figs 6&7- indicate diamide concentration. Where is data to show metabolism involvement as stated on p.13?

Response: In response to the comment of the Reviewer, we have indicated the diamide concentration in the legends of Figure 6 (page 27) and Figure 7 (page 28). The data showing differences in proteins related to metabolism are presented in Supplementary table 3, which is now indicated (page 13, line 445).

11. P. 14, top plus Fig. 8- no error bars in control plot to show real differences. What does slightly flatter curve mean?

Response:

To clarify what we meant with “a flatter curve”, we have rephrased the sentence. It now reads as follows: “..., whereas the *ΔribT* strain revealed a curve with a slightly less steep decline with more peptides in a higher oxidation state” (page 14, lines 483-485).

12. P. 14-paragraph 3- how many peptides for each protein were used to identify each class of proteins? Might include in Table 1.

Response: The prerequisite for protein identification was a minimum of two peptides, whereas for protein quantification, the prerequisite was to obtain at least two valid values from three

experimental replicates. This has now been specified in the Methods section on page 9 (lines 269-270 and lines 272-274).

13. P.14-15 "Notably" all these statements are not supported by data so why include?

Response: We appreciate this comment of the Reviewer, but actually all these data are presented in Supplementary table 3. To guide the reader to consult this table, we have now more regularly referred to it (see pages 14-15, lines 498, 505, 512, 524, 528, 533, 541 and 550).

14. P.16 7 lines up- what is the significance of the ribosomal protein changes? Is this reflected in rRNA as well?

Response: The Reviewer raises an interesting question, but in our present analysis we did not investigate RNA levels. RNA analyses would certainly represent an interesting follow-up of our present investigation. However, we believe that such analyses are beyond the scope of our present study, which was aimed at developing a diamide-based screening method for *Bacillus* strains with improved oxidative stress tolerance (see page 5, lines 101-102).

15. P.19 line 4- what does "variable" mean and why?

Response: We agree with the Reviewer that the word "variable" was a bit vague in the present context and have rephrased the sentence. It now reads as follows: "The encountered phenotypes could be traced back to different genomic disruptions and the mechanisms behind these phenotypes were also distinct" (page 19, lines 672-674).

16. Introduction- explain "more robust" and cite evidence.

Response: What we meant to say with the words "more robust" is that our present screening method will enhance the development of industrial production strains with improved robustness under conditions of oxidative stress. Of note, we used the wording "more robust" only in the Importance paragraph of our manuscript. We have rephrased the respective sentence to clarify what we mean (page 3, lines 53-54).

17. P.8 details in paragraphs 2&3 can be shortened.

Response: In principle, we could shorten these paragraphs. However, we believe that it will be helpful for the readers of our manuscript to keep these paragraphs as they are in the present manuscript version. This will help the readers to appreciate at a glance what we have done exactly. We therefore prefer not to shorten the two paragraphs. In case the Editor prefers shortening, we shall of course be ready to do this.

18. Do other oxidizing agents cause similar changes in designated genes or are they unique to diamide? Not clear what the value of resistant strains is and is there data for this? Would be valuable information to include.

Response: The Reviewer raises a relevant point that we actually addressed in the Introduction of our manuscript where we wrote the following: "Oxidative stress can originate from a variety of sources and is caused by oxidative damage of cellular components [10]. An example of this is the oxidation of free thiol groups of cysteine residues in proteins [11]. Molecules that can cause this type of stress are reactive oxygen, nitrogen, chlorine, and electrophilic species. The oxidative stress response generally refers to the adaptations made by the cell to deal with the adverse effects caused by these oxidants

[12]. The specific bacterial responses differ depending on the stress agent involved although overlap exists [10, 12]" (page 4, lines 67-73). The precise value of strains with improved resistance to oxidative stress still needs to be evaluated.

August 30, 2023

Dr. Michael Seefried
AB Enzymes GmbH
Feldbergstrasse 78
Darmstadt 64293
Germany

Re: Spectrum01608-23R1 (Diamide-based screening method for isolation of improved oxidative stress tolerance phenotypes in *Bacillus* mutant libraries.)

Dear Dr. Michael Seefried:

Your manuscript has been accepted, and I am forwarding it to the ASM Journals Department for publication. You will be notified when your proofs are ready to be viewed.

Sincerely,

Sacha Pidot
Editor, Microbiology Spectrum
